# Scattering of Microwaves by a Passive Array Antenna Based on Amorphous Ferromagnetic Microwires for Wireless Sensors with Biomedical Applications

**DOI:** 10.3390/s19143060

**Published:** 2019-07-11

**Authors:** Alberto Moya, Diego Archilla, Elena Navarro, Antonio Hernando, Pilar Marín

**Affiliations:** 1Instituto de Magnetismo Aplicado, Universidad Complutense de Madrid-ADIF-CSIC, P.O. Box, 155, Las Rozas, 28230 Madrid, Spain; 2Departamento de Física de Materiales, Universidad Complutense de Madrid, 28040 Madrid, Spain; 3IMDEA Nanociencia, C/Faraday 9, 28049 Madrid, Spain; 4Donostia International Physics Center (DIPC), 20018 San Sebastián, Spain

**Keywords:** magnetic, amorphous, microwires, wireless, biosensor, microwaves, giant magnetoimpedance, scattering

## Abstract

Co-based amorphous microwires presenting the giant magnetoimpedance effect are proposed as sensing elements for high sensitivity biosensors. In this work we report an experimental method for contactless detection of stress, temperature, and liquid concentration with application in medical sensors using the giant magnetoimpedance effect on microwires in the GHz range. The method is based on the scattering of electromagnetic microwaves by FeCoSiB amorphous metallic microwires. A modulation of the scattering parameter is achieved by applying a magnetic bias field that tunes the magnetic permeability of the ferromagnetic microwires. We demonstrate that the OFF/ON switching of the bias activates or cancels the amorphous ferromagnetic microwires (AFMW) antenna behavior. We show the advantages of measuring the performing time dependent frequency sweeps. In this case, the AC-bias modulation of the scattering coefficient versus frequency may be clearly appreciated. Furthermore, this modulation is enhanced by using arrays of microwires with an increasing number of individual microwires according to the antenna radiation theory. Transmission spectra show significant changes in the range of 3 dB for a relatively weak magnetic field of 15 Oe. A demonstration of the possibilities of the method for biomedical applications is shown by means of wireless temperature detector from 0 to 100 °C.

## 1. Introduction

Since the first publications on wire-based biosensor prototypes [1], many researchers have focused their efforts to develop soft magnetic materials and to understand its properties under the point of view of technological applications [2]. In particular, amorphous ferromagnetic microwires (AFMW) exhibiting giant magnetoimpedance (GMI) [3,4], bistability, and ferromagnetic resonance [5,6], are shown to be ideal candidates for technological [7,8,9,10,11] and medical applications [12,13], such as wireless pressure sensors [14], cardio-vascular tests [15], or structural sensors to inspect, for instance, the integrity of cardiovascular implants [16].

In addition, the tiny dimensions of AFMW in combination with their outstanding magnetic properties make them suitable for application on magnetic sensors [17,18], especially their GMI behavior has been of much research interest for many years [19,20]. GMI is a physical effect that expresses the large variation in the electrical impedance that occurs in some materials, in particular AFMW when subjected to an external magnetic field.

On the other hand, today a wide variety of modern technologies heavily depend on the interaction between ferromagnetic materials and microwave radiation [21,22,23]. During almost two decades, a huge amount of research has been devoted to the possibilities of controlling light propagation in microstructured media. For example, ferromagnetic resonance in the microwave range has been used to develop band-stop and notch filters [24]. Recently [25,26,27,28,29], some research has been oriented to understand the effect of GMI on the scattering of microwaves by a single AFMW. These works focus on both methodological and basic development and report experimental evidences of the dependence of microwave scattering by a single microwire on its magnetic permeability; an effect that is strong enough to be labeled as an effect of the GMI. Some of these works have shown different approaches by means of microwires forming arrays or buried in different types of matrix, searching to enhance their sensitivity as a GMI [30,31,32].

In this context Co-based microwires with circular magnetization in the surface region are an ideal scattering center for microwave radiation since they exhibit a noticeably high GMI effect [33]. The surface impedance is very sensible. The surface impedance of AFMW with circumferential anisotropy is very sensitive to the magnetic field even in the range of the GHz [34]. In addition, these Co rich amorphous alloys present negative low magnetostriction and therefore the scattering is also sensitive to mechanical stress. This fact has been profited for wireless stress sensing technologies [16] with very interesting possibilities of biomedical applications. A wireless system, based on this principle, has been developed for following up vascular surgery procedures. Magnetoelastic AFMW have demonstrated its capability for wireless detections of mechanical stress [35]. Magnetic microwires have been also use in a sensor system for early detection of heart valve bioprostheses failure [36].

In order to improve the detection sensitivity of magnetic microwires in the range of microwaves to be able to expand their detection possibilities in the field of biosensors, the present work analyzes and compares the microwave modulated scattering intensity produced by both, a single Fe_2.25_Co_72.75_Si_10_B_15_ amorphous microwire and, as the first time, an ensemble of them. A low frequency bias magnetic field tunes the permeability of AFMW. We have combined two effects: i) The improvement of the sensing capability based on microwave scattering by using ensemble of AFMWs; with ii) the advantages in measuring performing time dependent frequency sweeps where the modulation of the scattering parameter versus frequency may be appreciated if the sweep is done slowly enough. We show also that for appropriate distances between microwires in the arrangement, they behave like a passive ferromagnetic array antenna. We demonstrate that the OFF/ON switching of the DC-bias activates or cancels the antenna character of AFMWs. Finally, for the first time, we report contactless temperature measurements through microwire scattering.

## 2. Materials and Methods

AFMWs with nominal composition of Fe_2.25_Co_72.75_Si_10_B_15_ and different lengths between L = 6 and 12 cm were prepared by means of the modified Taylor-Ulitovsky technique [37]. Figure 1a shows scanning electron microscopy (SEM) images of a representative microwire with 49.7 μm diameter consisting of a metallic inner core with 31.4 μm diameter and a Pyrex cover. These AFMW present circumferential magnetization which results in an hysteresis loop with low coercivity and high permeability [38]. Figure 1b shows the room temperature hysteresis loop measured with an axial applied magnetic field in an AFMW with L = 0.5 cm by VSM magnetometer. The anisotropy field, $H_{K}$ estimated from Figure 1b is 10 Oe.

The microwave characterization of AFMW was carried out in the 500 MHz–8 GHz frequency range by placing the microwire between two horn emitting and receiving antennas with lineal polarized radiation. AGILENT E8362B Network Analyzer is connected to both antennas. The microwire is situated at 36 cm from both antennas parallel to the electric field of the microwave and orthogonal to the propagation direction. The distance between antennas was large enough to ensure the far field contribution dominates the near field one. Cu microwires with different lengths and 100 μm diameters were also characterized at a high frequency in order to be used as reference. 

Such an experimental arrangement allowed us to measure scattering coefficients of the system, $S_{ij}$, which are commonly used in telecommunications engineering to describe radio frequency systems. In particular, the transmission coefficient $S_{21}$ is defined as:(1)$$S_{21}=20\log_{10}\frac{P_{2}}{P_{1}}$$where $P_{1}\;$ is outgoing power in the emitting antenna and $P_{2}\;$ is the measured power in the receiving antenna. The scattering coefficient is directly related to the electrical current induced along the microwire by the incident electromagnetic wave since the scattered wave is generated by this electric current.

In order to determine the influence of the temperature on the scattering of the waves and the possibilities of the system in the wireless temperature measurement, the experiment was carried out in the time domain at a frequency of 2.1 GHz, on a microwire 6 cm in length, inside a glass capillary filled with water placed onside a heating element. Wireless and contact measurements have been simultaneously performed. Figure 2 shows the experimental set-up for frequency and time domain measurements including emitting and receiving antennas, Helmholtz coils and detail of magnetic microwire inside of a capillary on a heating resistance.

## 3. Results and Discussion

Figure 3a displays the spectra of the transmission coefficient $S_{21}\;$ in a frequency sweep between 0.5 and 5 GHz when a FeCoSiB AFMW or a Cu microwire respectively, with the same length, is placed between the antennas. The signal measured in the receiving antenna without microwires was subtracted before performing the transmission coefficients measurements. 

Let us first consider the two curves measured without applying the magnetic field. The frequency dispersion has a resonance character with a minimum in $S_{21}$ at the frequency that, as will be shown later, defines the operating frequency of the device. It is well known that the induced current in the wires is maximal under resonance conditions (dipole antenna resonance). The minimum in $S_{21}$ is located at the same frequency (~2.1 GHz) for both 6 cm length FeCoSiB and Cu microwires. This behavior is in accordance with the cylindrical antenna theory giving evidence of the antenna character for FeCoSiB AFMW. The difference in the $S_{21}\;$ minimum amplitude between two microwires, is due to the different resistivity of Fe_2.25_Co_72.75_Si_10_B_15_ and Cu wires (1.69 × 10^−8^ and 4.83 × 10^−8^ Ωm respectively).

As previously shown [14,27], the electrical current can be modified by changing the magnetic permeability, μ, of the AFMW. For this reason, two Helmholtz coils with a customized electronic set up, were used to apply both a DC and a low frequency (20 Hz) AC-magnetic bias fields. Both, DC and AC-bias have the same amplitude, ($H_{appl}=15$ Oe) being parallel to the microwire axis. $H_{appl}$ is higher than the anisotropy field, $H_{K}\;$ associated with circumferential anisotropy of the AFMW as it has been previously reported [29].

The electric current in the AMFW is computed by the Hallen-Pocklington equation [39]. The differential magnetic permeability of the AFMW, μ=δM/δH, determines the wire impedance per unit length, $Z_{i}$ according to equation the general expression: (2)$$Z_{i}=\sqrt{\frac{\omega \mu }{2\sigma }}\;\frac{\left(1-i\right)J_{0}\left(\frac{\left(1-i\right)a}{\delta }\right)}{2\pi aJ_{1}\left(\frac{\left(1-i\right)a}{\delta }\right)}$$where 2a is the characteristic cross-section, σ is the electrical conductivity of the AFMW, $J_{0}$ and $J_{1}$ are the first-kind Bessel functions, and $\delta =\sqrt{2/\omega \mu \sigma }$ is the magnetic skin depth. References [31,40] are good examples about the relationship between the magnetic permeability and the impedance of an AFMW. The induced current in the wire is modulated, through a bias field, by means of the magnetoimpedance effect. 

First of all, we have analyzed the effect on the microwave scattering of a DC-field applied parallel to the FeCoSiB microwire axis. In Figure 3a an almost constant behavior of $S_{21}$ vs. frequency is measured when an axial DC-bias of $H_{appl}=15\;\mathrm{Oe}>H_{K}$ is applied to the AFMW. The lack of absorption minimum between 500 MHz and 5 GHz means that an axial saturation state in AFMW with circumferential anisotropy cancels its antenna character making it transparent to electromagnetic radiation in that frequencies range.

According to theoretical predictions, transmission spectra have a very large sensitivity near the antenna resonance. At the resonant frequency, the current distribution in the wires strongly depends on their surface impedance. In general, the scattering properties of wires are greater if the surface impedance is low [41]. 

The surface impedance on Co-based AFMW is minimum when magnetization is along the azimuthal direction, which is the case for a circumferential anisotropy at zero axial applied field [31]. Therefore, a deep minimum in transmission is seen in the case of DC-bias OFF (Figure 3a). According to Equation (2), the surface impedance in the wire increases in the presence of the field as $\sqrt{\mu }$ since the magnetization rotates towards the wire axis. Consequently, the low and high surface impedances are switched by the axial magnetic field that turns the magnetization from circumferential to longitudinal orientation.

In addition, $Z_{i}(\mu )$, allows for modulation of the induced current in the wire through a low AC-magnetic field. The AC-field gives rise to a modulation in the transmission coefficient of microwaves (Δ$S_{21}$). Figure 3b shows the Δ$S_{21}$ modulation due to an AC-bias in a time dependent frequency sweep performed with a scanning speed slow enough to be sensible to the permeability modulation. This curve would be the “ON” prompt reply of our magnetic AC-field sensor element in contrast with the ”OFF” one represented by the zero Δ$S_{21}\;$ modulation (red horizontal line in Figure 3b). This Δ$S_{21}=0$ was obtained after subtracting the curve $S_{21}\;\left(f\right)$ for FeCoSiB with DC-bias OFF, shown in Figure 3a. Next, the AC-bias was applied and the AC-bias ON curve displayed in Figure 3b was measured. Therefore, in this second type of characterization, the highest values of Δ$S_{21}$ are measured each time than $H_{appl}$ reaches its maximum value. 

The effect of the AC-bias field on Δ$S_{21}\;$ modulation is very large near the resonance. In Figure 3b it can be seen that the modulation of the transmission coefficient is maximum at resonant frequency in correspondence with $S_{21}$ minimum shown in Figure 3a. The maximum variation of the induced current by changing the magnetic permeability corresponds to that obtained at the dipole antenna resonance. 

In order to analyze the modulation effect with more detail, the insert of Figure 3b shows a time domain measurement where $S_{21}$ is represented as a function of time fixing the frequency of emission antenna to the AFMW resonance value ($f_{antenna}=\;2.1\;\mathrm{GHz}$ for L = 6 cm). The modulation of the reflected power P_2_ at the receiving antenna originated by the low frequency bias field is described by the scattering parameter behaviour. When the applied field equals the anisotropy field of the AFMW the maximum permeability variation induced by the applied field is observed. If $H_{appl}>H_{K}$, the field modulation of the scattering decreases and finally disappears for fields well above the saturation field [25]. The frequency of the modulation ($f_{mod}=40\;\mathrm{Hz}$) is twice the frequency of the bias field one ($f_{bias}=20\;\mathrm{Hz}$). This result confirms that the scattering parameter is tuned by the permeability in the microwire and is a consequence of the symmetric shape of the hysteresis loop. Therefore, FeCoSiB microwires work as a contactless sensor capable of following the magnetic field, and sensitive to very low magnetic fields. 

The length of the microwires determines its scattering characteristics through the dipolar electromagnetic effect. As can be seen from Figure 3b–d, the maximal frequencies in the Δ$S_{21}$ modulations are 2.1, 1.5 and 1 GHz for L = 6, 9 y 12 cm respectively. 

The next step of this work was to try to increase the sensitivity of our field contactless sensor. It is well known that antenna arrays are an important class of antennas which are widely used in point to point communication systems, where a very high directive beam of radiation is needed. One, two, and four-element arrays of half-wave dipole antenna are shown in Figure 4a–c, respectively. The dipoles are in XZ plane parallel to Z-axis and separated λ/2 being λ the wavelength of the microwave radiation. The radiation field pattern of each array is shown next to it. For these radiation field patterns, the finite element simulation code COMSOL Multiphysics has been employed. As the number of dipoles increases, the radiation is concentrated on the Y-axis becoming more directional. If a still narrower beam is desired, additional elements can be added. 

In this context, as AFMWs behave as a dipolar antenna with the resonant at half wave length condition, an arrangement of several AFMWs between the emitting and the receiving antennas can enhance the sensibility of our magnetic field detector. Figure 5 shows the microwave characterization for the three different cases shown in Figure 4: Figure 5a for 1 AFMW, Figure 5b for 2 AFMWs, and Figure 5c for 4 AFMWs.

The microwires behave as a dipolar antenna with the resonant at half wave length condition: $f_{r}=\frac{c}{2L}$ where L is the wire length, c is the velocity of light, and ε the relative permeability. All of the microwires have a length of 6 cm (L~λ/2), and the separation between them was kept to 6 cm (d~λ/2). Figure 5 shows that even though the resonance frequency does not depend on the number of microwires, an enhancement of the Δ$S_{21}$ amplitude modulation (up to 3 dB) with the number of microwires is observed. The increase of Δ$S_{21}$ achieved with a small magnetic field can be easily improved with the number of AFMW in the array.

An interesting example of the wireless detection capability of the physical parameters of this method is the experiment shown in Figure 2. A magnetic microwire of 6 cm in length, located inside a capillary filled with water, is located on top of a heating resistance through which a controlled current intensity circulates. The temperature is determined by a thermocouple located in the vicinity of the wire. The experiment is performed in the presence of a 2.1 GHz wave corresponding to the microwire antenna resonance frequency and an axial AC-bias of $H_{\mathrm{appl}}=15\;\mathrm{Oe}$. Time domain measurements of parameter S_21_ are made for different temperatures under the same conditions as the graph shown in the insert of Figure 3b. Figure 6a shows how the temperature influences the modulation of the signal by making the depth of the detection peak smaller as the temperature increases. Figure 6b depicts the relationship between the value of the temperature that has been measured using a thermocouple that collected wirelessly using a detector antenna placed at a distance of 50 cm. Temperature variations of ΔT~80 K can be registered as relative changes on the ΔS_21_ peak of about 18%. In accordance with Reference [42], the maximal sensitivity of the impedance to the external magnetic field was observed at minimal temperature.

The observed evolution of Δ$S_{21}$, with temperature increase, should be ascribed to the combination between the increase of the electrical resistance of the metal as well as the evolution of the circular anisotropy. An increase in temperature supposes material stress relaxation and, due to negative magnetostriction constant, the decrease in mechanical stress supposes a decrease on circular magnetic anisotropy, intrinsically related to microwire impedance by means of magnetic permeability. The higher the impedance, the lower the current in the wire and therefore the less scattering and the less variation of the parameter S_21_. Previous works [43] have reported temperature dependence of GMI in amorphous microwires for sensor application but in that cases the measures were performed with an excitation current using electric contacts.

## 4. Conclusions

In this work we have modulated microwaves by means of AFMW arrays. The length of the wires should be of the order of the wavelength to fulfill the antenna resonance condition. The observed effect is a consequence of the large magnetoimpedance at GHz frequencies of AFMW: The antenna resonance condition is influenced by applying a weak magnetic field. Microwave scattered intensity has been controlled by both, tuning its permeability and the number of microwires in the array. The achieved tunability reaches 3 dB for magnetic applied fields of 15 Oe. We showed that switching OFF/ON the bias, activates or cancels the AFMW antenna behavior. These effects can be used for different sensing applications like wireless field sensor as demonstrated in the experiment performed as a function of temperature. It is shown that there is a linear correlation between the scattering parameter Δ$S_{21}$ and the temperature, so this method can be considered as an alternative for wireless temperature measurement.

## Figures and Tables

**Figure 1 sensors-19-03060-f001:**
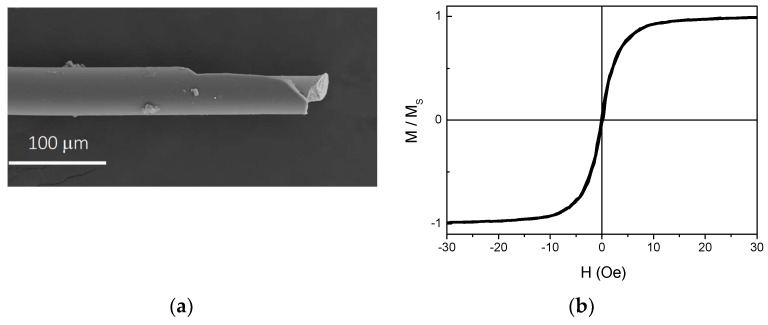
(**a**) SEM general view of a representative microwire consisting of a metallic inner core and a Pyrex cover. (**b**) Room temperature hysteresis loops of a L = 0.5 cm amorphous ferromagnetic microwires with an axial applied magnetic field.

**Figure 2 sensors-19-03060-f002:**
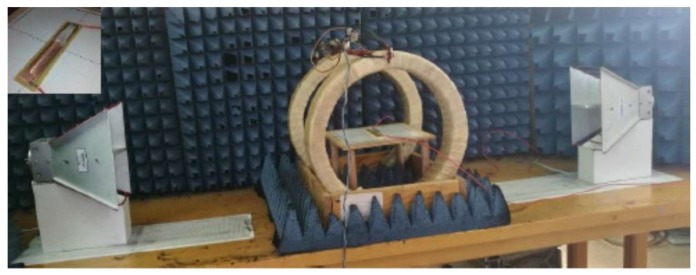
Experimental set-up for frequency and time domain measurements including emitting and receiving antennas, Helmholtz coils and detail of magnetic microwire inside of a capillary on a heating resistance. Insert: Detail of magnetic microwire inside of a capillary on a heating resistance.

**Figure 3 sensors-19-03060-f003:**
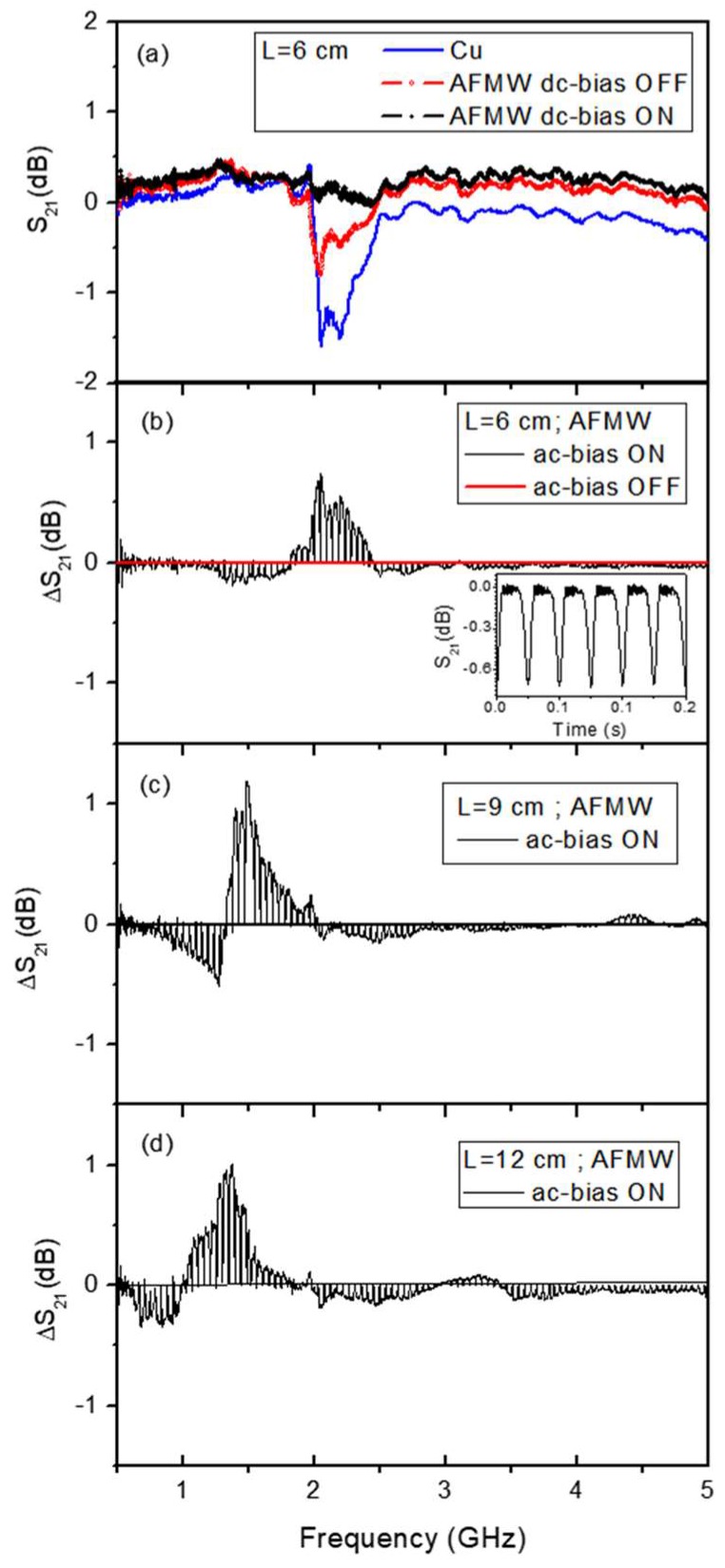
(**a**) Spectra of the transmission coefficient ($S_{21}$ parameter) in a frequency sweep between 0.5 and 5 GHz for a 6 cm length Cu microwire and a 6 cm length FeCoSiB microwire without bias field. (**b**) Modulation of the scattering coefficient, Δ$S_{21}$, due to an AC-bias magnetic field with an amplitude of 15 Oe and a frequency of 20 Hz (solid black line). The horizontal red line represents the situation without magnetic field. This line is obtained after subtracting the signal in Figure 3a. The insert shows a $S_{21}$ time domain measurement at 2.1 GHz in FeCoSiB microwire with L = 6 cm. (**c**,**d**) Δ$S_{21}$ modulation versus frequency measured with identical conditions than (a) for 9 cm and 12 cm length, respectively, FeCoSiB wires.

**Figure 4 sensors-19-03060-f004:**
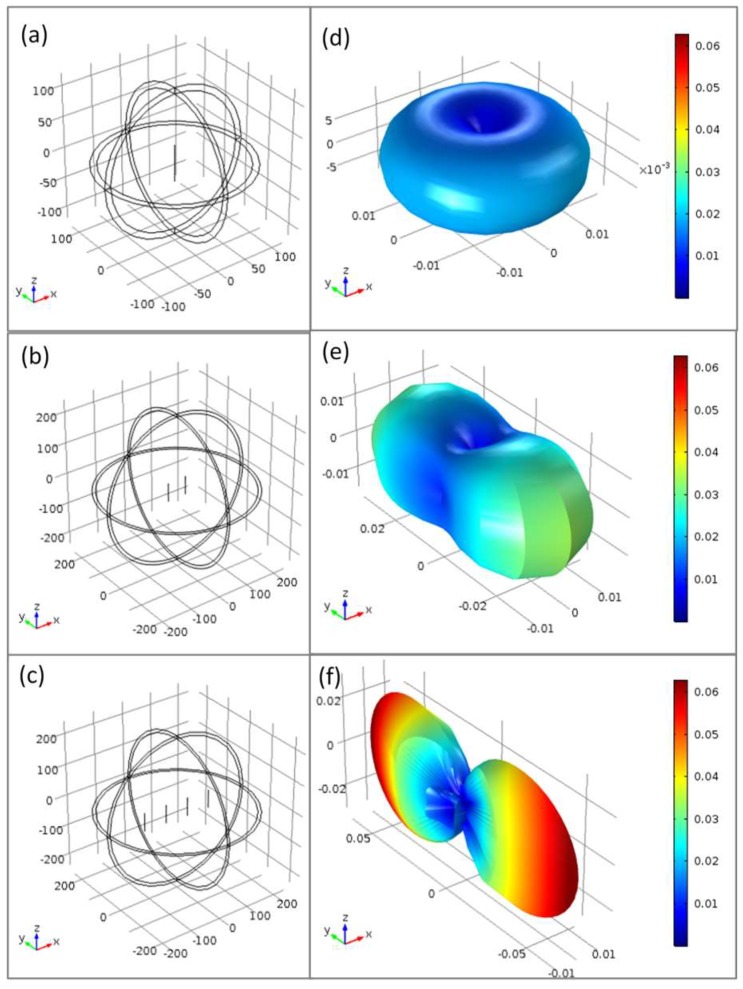
(**a**–**c**): One, two and four-element arrays of half-wave dipole antenna respectively separated λ/2. (**d**–**f**): COMSOL simulations of radiation field patterns of each array having a half-wavelength spacing between dipoles with I = 0.4 mA.

**Figure 5 sensors-19-03060-f005:**
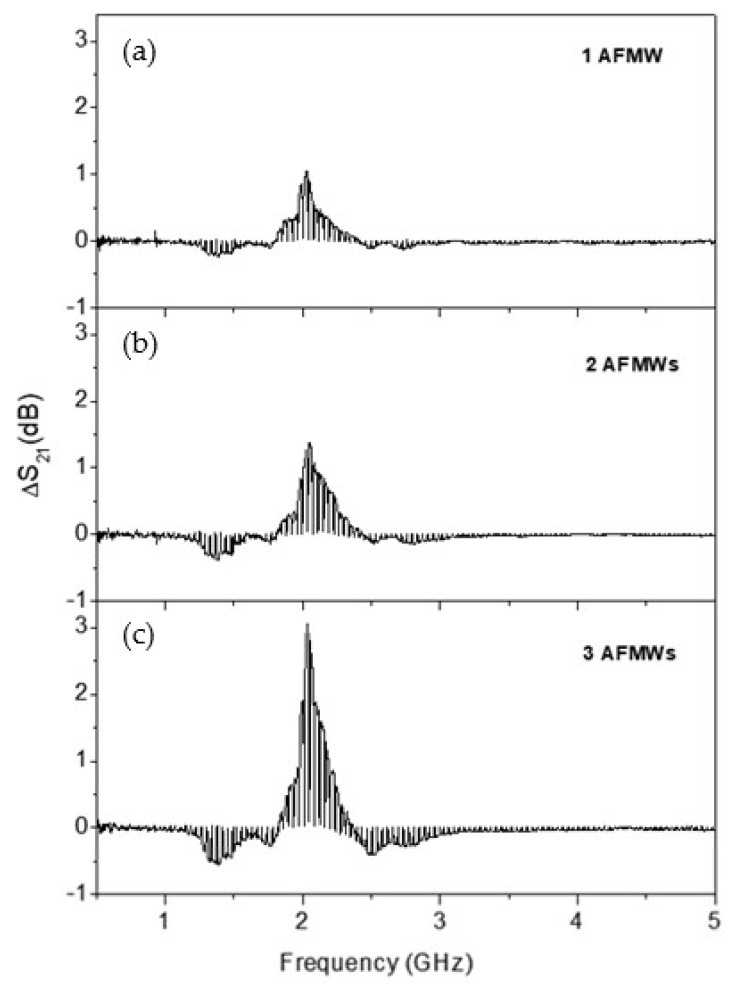
Modulation of the scattering coefficient, Δ$S_{21}$ versus frequency for three different cases: (**a**) 1 AFMW, (**b**) 2 AFMWs and (**c**) 4 AFMWs. All of them have 6 cm length and the separation was kept to 6 cm.

**Figure 6 sensors-19-03060-f006:**
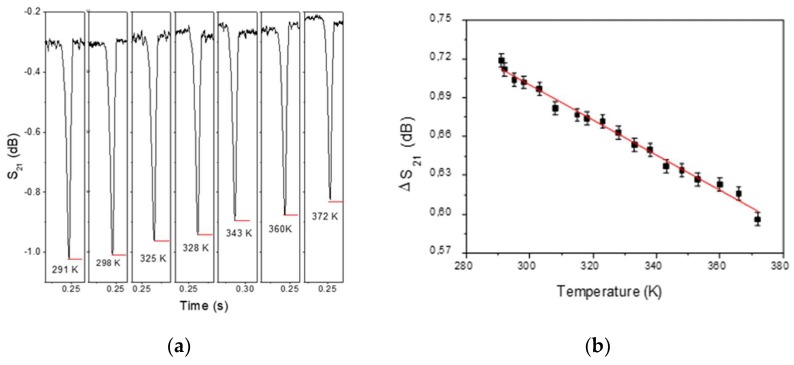
(**a**) Evolution of domain time scattering signal S_12_ with measuring temperature for 6 cm length microwire at 2.1 GHZ. (**b**) Linear evolution of peak intensity, Δ$S_{21}$, with measuring temperature.

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
