# Peer review of "Scattering of Microwaves by a Passive Array Antenna Based on Amorphous Ferromagnetic Microwires for Wireless Sensors with Biomedical Applications"

_sensors, 2019, doi:10.3390/s19143060_

Round 1
Reviewer 1 Report
The paper deals with the microwave tunability properties of Co-based amorphous microwires, single or in arrays. The paper is well organized and the results are clearly presented. However, it would be beneficial for the readers if the authors could provide a few more details concerning the novelty aspects of the present work compared with their previous work. Also, please explain how are the presented results connected with the title saying "wireless sensors with biomedical applications". This connection is not obvious from the text. Otherwise, the paper is suitable for publication.
Author Response
Response 1: The information required by the referee can be found in the last paragraph of the introduction section. However, more details have been included in order to clarify the novelty aspect of the present work compared with our previous results and the connection with the title:
“In order to improve the detection sensitivity of magnetic microwires in the range of microwaves to be able to expand their detection possibilities in the field of biosensors the present work analyzes and compares the microwave modulated scattering intensity produced by both, a single Fe2.25Co72.75Si10B15 amorphous microwire and, as the first time, an ensemble of them. The modulation is driven by applying a bias magnetic field able to tune the permeability of the ferromagnetic microwire. We have combined two effects: i) the improvement of the sensing capability based on microwave scattering by using ensemble of AFMWs; with ii) the advantages in measuring performing time dependent frequency sweeps where the modulation of the scattering parameter versus frequency may be appreciated if the sweep is done slowly enough. We show also that for appropriate distances between microwires in the arrangement, they behave like a passive ferromagnetic array antenna. We demonstrate that the OFF/ON switching of the dc-bias activates or cancels the antenna character of AFMWs. Finally, for the first time, we report contactless temperature measurements through microwire scattering.”
Reviewer 2 Report
Submitted manuscript is interesting. The main goal and results are well suiting the Sensors journal but it might get even higher visibility and citation impact in particularly specialized issue like "Biosensors with Magnetic Nanocomponents". Despite sufficient novelty of the content, some points certainly require revision prior to possible publication.
1. Magnetic impedance in amorphous wires was studied for a long time and at least one of classic MI works on this subject must be cited (for example Beach, R.S. and Berkowitz, A.E., Giant Magnetic Field Dependent Impedance of Amorphous FeCoSiB Wire, Appl. Phys. Lett., 1994, vol. 64, pp. 3652–3654; Paramonov, V.P., Antonov, A.S., Lagarikov, A.N., Panina, L.V., and Mohri, K., High Frequency (1–1200 MHz) Magnetoimpedance in CoFeSiB Amorphous Wires, J. Appl. Phys., 1996, vol. 79, p. 6532).
2. As magnetic biosensing with the wires is considered Authors also could consider very first publications on wire-based biosensor prototypes (Chiriac, H.; Herea, D.-D.; Corodeanu, S. Microwire array for giant magnetoimpedance detection of magnetic particles for biosensor prototype. J. Magn. Magn. Matter. 2007, 311, 425–428.).
3. For magnetic bistability, low field absorption and FMR some classic references would be also an advantage (A. Yelon, D. Menard, M. Britel, and P. Ciureanu, Calculations of Giant Magnetoimpedance and of Ferromagnetic Resonance Response are Rigorously Equivalent, Appl. Phys. Lett. 69, 3084–3085 (1996); N. Buznikov, The Effect of Surface Domain Structure on Low-Field Microwave Absorption of Magnetic Microwires, J. Phys. D: Appl. Phys. 43, 055002 (2010)).
4. Recently MI sensor element was proposed for cardio-vascular tests and it might be also used for the discussion (Felix A. Blyakhman, Emilia B. Makarova, Fedor A. Fadeyev, Daiana V. Lugovets, Alexander P. Safronov, Pavel A. Shabadrov, Tatyana F. Shklyar, Grigory Yu. Melnikov, Iñaki Orue, Galina V. Kurlyandskaya, The Contribution of Magnetic Nanoparticles to Ferrogel Biophysical Properties Nanomaterials 2019, 9, 232).
5. As the work devoted to the very high frequency applications recent advances in the field must be mentioned both for methodological and basis development (G.V. Kurlyandskaya, S.V. Shcherbinin, S.O. Volchkov, S.M. Bhagat, E. Calle, R. Pérez, M. Vazquez, Soft magnetic materials for sensor applications in the high frequency range; Journal of Magnetism and Magnetic Materials 459 (2018) 154–158; Chen, J.; Li, J.; Li, Y.; Chen, Y.; Xu, L. Design and fabrication of a miniaturized GMI magnetic sensor based on amorphous wire by MEMS technology. Sensors 2018, 18, 732 ).
6. Thermal effects and MI in wires were studied by different groups and their contributions can be also appreciated and discussed (Semirov, A.V.; Derevyanko, M.S.; Bukreev, D.A.; Moiseev, A.A.; Kudryavtsev, V.O.; Safronov, A.P. Magnetoimpedance of cobalt-based amorphous ribbons/polymer composites. J. Magn. Magn. Mater. 2016, 415, 97–101; Nabias, J.; Asfour, A.; Yonnet, J.-P. Temperature dependence of giant magnetoimpedance in amorphous microwires for sensor application. IEEE Trans. Magn. 2017, 53, 4001005).
7. Minimum magnetic characterization (at least room temperature hysteresis loop) must be given as well as SEM image would be an advantage.
8. References have some misprints (see Ref. 3 as an example).
Round 2
Reviewer 2 Report
Authors made very good revision and responded all my questions. Work now can be accepted in the present state.